# A Quantitative Approach to Potency Testing for Chimeric Antigen Receptor-Encoding Lentiviral Vectors and Autologous CAR-T Cell Products, Using Flow Cytometry

**DOI:** 10.3390/pharmaceutics17030303

**Published:** 2025-02-25

**Authors:** Juan José Mata-Molanes, Leticia Alserawan, Carolina España, Carla Guijarro, Ana López-Pecino, Hugo Calderón, Ane Altuna, Lorena Pérez-Amill, Nela Klein-González, Carlos Fernández de Larrea, Europa Azucena González-Navarro, Julio Delgado, Manel Juan, Maria Castella

**Affiliations:** 1Department of Immunology, CDB, Hospital Clinic de Barcelona (HCB), 08036 Barcelona, Spain; jjmata@clinic.cat (J.J.M.-M.); alserawan@clinic.cat (L.A.); cespana@clinic.cat (C.E.); cguijarro@clinic.cat (C.G.); hcalderon@clinic.cat (H.C.); eagonzal@clinic.cat (E.A.G.-N.); mjuan@clinic.cat (M.J.); 2Immunogenetics and Immunotherapy in Autoinflammatory and Immune Responses Group, Fundació Clínic-IDIBAPS, 08036 Barcelona, Spain; lopez72@recerca.clinic.cat (A.L.-P.); altuna@recerca.clinic.cat (A.A.); loperez@recerca.clinic.cat (L.P.-A.); klein@recerca.clinic.cat (N.K.-G.); 3School of Medicine, Universitat de Barcelona (UB), 08036 Barcelona, Spain; cfernan1@clinic.cat (C.F.d.L.); jdelgado@clinic.cat (J.D.); 4Department of Hematology, Institute of Cancer and Blood Diseases, Hospital Clinic de Barcelona (HCB), 08036 Barcelona, Spain; 5Fundació Clínic-IDIBAPS, 08036 Barcelona, Spain

**Keywords:** chimeric antigen receptor, potency testing, T-cell activation, analytical method validation, ATMP development

## Abstract

**Background/Objectives**: Potency testing of clinical-grade lentiviral vectors (LVVs) is critical to support a drug’s commercial approval. Careful consideration should be paid to the development of a suitable potency test during the drug’s clinical development. We aimed to develop an affordable, quantitative test for our CAR19-LVV, based on a measure of transgene’s functional activity. **Methods**: Several indicators of functional activity of CAR19-LVV were explored in a co-culture setting of CAR-transduced Jurkat cells and CD19-expressing target cells. The selected assay was further developed and subjected to validation. Assay’s adaptability to other CAR-encoding LVV and autologous CAR-T cell products was also investigated. **Results**: Measure of CD69 expression on the membrane of Jurkat-CAR-expressing cells is a specific indicator of CAR functionality. Quantification of CD69 in terms of mean fluorescence intensity (MFI), coupled with an intra-assay standard curve calibration, allows for a quantitative assay with high precision, specificity, robustness, linearity and accuracy. The assay has also shown optimal performance for a CARBCMA-LVV product. Importantly, we show that in primary T cells, CD69 expression reflects CAR-T cell cytotoxicity. After adaptation, we have applied a CD69-based potency test, with simultaneous measurement of CAR-T cell cytotoxicity, to autologous CAR-T cell products, demonstrating the assay’s specificity also in this context. **Conclusions**: We developed a validated, in vitro cell-based potency test, using a quantitative flow-cytometry method, for our CAR19-LVV. The assay is based on the detection of T-cell activation upon CAR binding to antigen, which is a measure of transgene functionality. The assay was easily adapted to another CAR-encoding LVV, targeting a different molecule. Furthermore, the same assay principle can be applied in the context of autologous CAR-T cell products. The quantitative CD69 potency assay shows reduced variability among autologous products compared to the IFNγ assay and allows for simultaneous evaluation of traditional semi-quantitative cytotoxicity, thereby directly evaluating the drug’s mechanism of action (MoA) in the same assay.

## 1. Introduction

Lentiviral vectors (LVVs) are widely used as a transgene delivery system during the manufacturing of advanced therapeutic medicinal products (ATMPs), especially in the fields of immunotherapy for cancer and for the treatment of monogenic disorders [1,2,3].

The number of LVV batches produced worldwide has been increasing exponentially for the last 10 years, due to the increasing number of clinical trials and approved products that include this technology [3]. As ATMPs transition towards later phases of clinical development, the requirement to have a quantitative potency test in place for the release of LVV batches becomes critical to obtain commercial approval by regulatory agencies [4]. The same applies to CAR-T cells and other gene therapy products. As stated in the EMA guideline for potency testing of cell-based immunotherapy products for cancer treatment [5], potency is the quantitative measure of biological activity based on the attribute of the product, which is linked to its mechanism of action (MoA).

CAR-T cells are a type of adoptive T-cell therapy (ACT) based on genetically modified T cells that express a CAR on the cell surface. The CAR molecule displays affinity towards a tumor-specific antigen, thus, redirecting T-cell cytotoxic activity towards cancer cells. Upon antigen binding, T cells become activated, resulting in active T-cell proliferation and release of granzymes and other cytotoxic molecules, finally resulting in tumor-cell lysis [6,7,8,9]. CAR T-cells for the treatment of several hematologic malignancies, including acute or chronic lymphocytic leukemias (ALLs or CLLs), several types of non-Hodgkin’s lymphomas (NHLs) and multiple myeloma (MM), showed unprecedented efficacy rates in refractory/relapsed (R/R) patients [10,11,12,13,14,15,16]. Thus, the field of CAR-T cells is continuously expanding, with new applications also for solid tumors, autoimmune diseases and infectious diseases [17,18,19].

ARI-0001 is an academically developed CAR-T cell product targeting CD19 antigen for the treatment of B-cell malignancies [20,21]. In an initial pilot study (EudraCT 2016-002972-29), ARI-0001 demonstrated a good safety and efficacy profile in adult and pediatric patients of ALL, NHL and CLL. These results are currently being further evaluated in phase II clinical trials (EudraCT 2019-003038-17, 2022-001101-52) for the indication of adult and pediatric B-ALL, respectively.

Several types of potency tests for *CAR*-encoding LVVs and/or CAR-T cell products have been proposed by both academic and industry sectors [22,23,24,25]. CAR-T cell’s MoA consists of the lysis of the tumor cells within the patient’s body. During the pre-clinical development of CAR-T cell products, cell-killing assays, performed both in vitro and in vivo, are fundamental tools for the evaluation of both the efficacy and safety of the product [26]. However, such assays are not necessarily ideal for routine quality control (QC) potency testing of the product due to the intrinsic assay variability and difficulties in assay transferability between laboratories, in part, due to the difficulty in establishing internal standards for the assay. Moreover, the replacement of in vivo assays with adequate in vitro assays for potency testing is highly encouraged by regulatory agencies, following the principles of the 3Rs. For these reasons, regulatory agencies allow the use of “surrogate” potency assays, closely related to the drug’s MoA, that are more suitable for QC routine testing [5]. These assays include measurement of degranulation markers (CD107a or granzyme B), proinflammatory cytokines (IFNγ, TNFα or IL-2), or T-cell activation markers (CD69, CD25, HLA-DR) [23,25].

Here, we developed an alternative potency test for ARI-0001 lentiviral vector (CAR19-LVV), using a quantitative flow cytometry-based method. The assay relies on the monitoring of specific T-cell activation, by means of CD69 increased expression that occurs upon interaction of the CAR molecule with its ligand, as part of the T-cell cytotoxic response [27]. We show that the assay was successfully validated in terms of precision, linearity and range, specificity and relative accuracy. We also show that CD69 is a valid “surrogate” potency test as it shows a correlation with T-cell cytotoxicity and explores a potential implementation of the assay for routing QC testing of autologous CAR-T cell products.

## 2. Materials and Methods

### 2.1. Cell Lines, Patient Samples and Culture Conditions

Jurkat and NALM6 cell lines were obtained from the DSMZ collection. U266 cells were obtained from ATCC. NALM6-CD19 clones with various levels of CD19 expression were kindly provided by Dr. Crystal L. Mackall [28]. NALM6-CD19-KO cell line was generated using the CRISPR/Cas9 system as follows: Two sgRNA targeting exon 1 and exon 2 were used. RNP complexes were prepared with Cas9 protein (TrueCut Cas9 v2 Protein, ThermoFisher) at a 3:1 ratio and incubated at room temperature for 10 min. Cells were then electroporated using Lonza 4D-Nucelofector (Lonza, Basel, Switzerland) using the manufacturer’s recommended program for this cell line. On the fifth day, knockout efficiency was evaluated by flow cytometry, and the CD19-negative population was isolated via fluorescence-activated cell sorting (FACS). All cell lines were cultured with RPMI supplemented with 10% FBS and 1% penicillin–streptomycin and incubated at 37 °C, 5% CO_2_.

Untransduced and CAR19-expressing primary T cells from patients included in clinical trials EudraCT 2016-002972-29, 2019-003038-17 or 2022-001101-52 were included in this study. This study has been approved by the Research Ethics Committee (CEIm) of Hospital Clinic: HCB/2017/0001 (Clinical trial: CART19-BE-01, Eudra: 2016-002972-29) and HCB/2019/0818 (EudraCT 2019-003038-17). It has also received approval from the Research Ethics Committee (CEIm) of Hospital Sant Joan de Déu; AC-15-22 (EudraCT 2022-001101-52).

### 2.2. Antibodies and Flow Cytometry

For flow cytometry analysis, cells were washed with PBS and centrifuged at 300 g for 5 min. Cells were then resuspended in residual volume and incubated with the indicated antibodies for 15 min at room temperature in the dark. After incubation, cells were washed with PBS and acquired using an Attune NxT Flow Cytometer (ThermoFisher, Waltham, MA, USA). Flow cytometry data were analyzed using FlowJo v10 software.

Monoclonal antibodies used were as follows: anti-CD25-PE (BD, Cat. N. 560989), anti-CD69-APC-Cy7 (BD, Cat. N. 557756), anti-CD69-APC (BD, Cat. N. 555533), anti-CD69 blocking antibody (Miltenyi, Cat. N. 130-124-326), anti-CD3-BV421 (BD, Cat. N. 563798) and anti-CD19-PE (BD, Cat. N. 555413). CAR19 was detected using APC-conjugated AffiniPureF(ab’)_2_ Fragment Goat anti-mouse IgG (Jackson Immunoresearch, Cat. N. 115-136-072). 7-AAD was used to assess cell viability (BD, Cat N. 559925). For assays, where cytotoxicity was also analyzed, a fixed volume was acquired with a flow cytometer, and absolute cell counts were recorded for analysis.

Quantum™ APC MESF beads (Bangs laboratories, Cat. N. 823) were used as an internal calibrator and run in parallel within each experiment. For fluorochrome–protein ratio calculation, Quantum™ Simply Cellular^®^ anti-Mouse IgG (Bangs laboratories, Cat. N. 815) was used following the manufacturer’s instructions.

To quantify CD19 molecules expressed on the cell’s surface of NALM6 cell clones, the Quantibrite™ PE Phycoerythrin Fluorescence Quantitation Kit (BD, Cat. N. 340495) was used following the manufacturer’s instructions.

### 2.3. Jurkat Cell Transduction

5 × 10^5^ Jurkat cells were seeded in 2 mL of RPMI complete media in 6-well plates. Cells were left untransduced or CAR19-LVV was added at the indicated MOI during the tests conducted in the development phase. For the established potency assay, CAR19-LVV is added at different MOIs, ranging from 0.25 to 4. Cells were incubated for 72 h. After this time, 100 µL of the sample from each well was separated and processed for CAR19 detection using flow cytometry.

### 2.4. Co-Culture (E:T Cells) Experiments Using Jurkat Cells

Effector (Jurkat cells—untransduced or CAR19-transduced) and target (NALM6) cell co-cultures were established in 96-well round-bottom plates in a final volume of 150 µL per well. Each condition was performed in triplicate wells. Effector cells were seeded at a constant cell number of 40,000 cells/well. Number of target cells was calculated according to the indicated E:T cell ratio. Co-cultures were incubated for 24 h unless otherwise indicated. After incubation, cells were centrifuged at 300 g for 5 min. The supernatant was recovered and stored for cytokine measurement when necessary. The cell pellet was resuspended in PBS and processed for flow cytometry analysis.

### 2.5. Co-Culture (E:T Cells) Experiments Using Primary T Cells

The CAR-T cell production method is described in Castella et al. 2020 [20]. Briefly, T cells were manufactured using the CliniMACS Prodigy system. T cells were selected from the patient’s apheresis using TransACT (Miltenyi, Cat.N. 200-076-202). Then, T cells were cultured in TexMACS medium (Miltenyi, Cat.N. 170-076-306) + 3%AB Serum for 24 h before transduction with CAR19-LVV. After transduction, cells were cultured for 7–9 days before cryopreservation.

At the time of potency testing, CAR-T cells (or Untransduced T cells) were thawed and cultured in XVivo15 media + 5%AB serum + 1%P/S + 50 U/mL IL-2 for 24 h before plating cells for potency-assay tests.

Co-cultures were established in 96-well round-bottom plates in a final volume of 250 µL per well using XVivo15-supplemented media (*w*/*o* IL-2). Each condition was performed in triplicate wells. Effector cells were seeded at a constant cell number of 4 × 10^5^ CAR+ cells/well. 8 × 10^5^ target cells (NALM6) were added to wells corresponding to E:T ratio = 1:2. Co-cultures were incubated for 24 h. After incubation, cells were centrifuged at 300 g for 5 min. The supernatant was removed, and the cell pellet was resuspended in PBS and processed for flow cytometry analysis.

### 2.6. IL-2 and IFNγ ELISA

Secreted IL-2 and IFNγ were measured in the supernatants of co-cultures using a Human IL-2 quantikine ELISA kit (R&D, Cat. N D2050) and DuoSet Human IFNγ ELISA kit (R&D, Cat. N DY285B-05), respectively. The assay was performed according to the manufacturer’s instructions. The supernatant was collected for 3 replicate wells for each condition. Absorbance was quantified using the BioTek Epoch instrument (Agilent, Santa Clara, CA, USA) and analyzed using Gen5 software (v3.10).

### 2.7. Statistics

Statistics were performed using Prism GraphPad software (v8.0.1). For comparisons of two sets of normally distributed data, a student’s *t*-test was used. For linear correlations between two sets of data, Pearson’s correlation was used. Non-linear regression (curve fit) was also used when applicable.

## 3. Results

### 3.1. Selection of a Potency Assay for ARI-0001 LV Vector

We aimed to develop a suitable potency assay to assess the transduction and transgene functionality of CAR19-LVV. For this, a cell-based assay was required. We selected Jurkat cells, a T-ALL-derived cell line, as a representative effector cell. Jurkat cells lack significant cytolytic activity; however, CD69 and CD25 upregulation and IL-2 secretion upon activation have been reported [29,30]. As a target cell line, the NALM6 cell line, which expresses CD19 and has been previously shown to activate ARI-0001 cells (primary T cells transduced with CAR19-LVV), was selected [20,21].

As a first screen, co-cultures of untransduced (UNT) or CAR19-transduced (CAR) Jurkat cells with NALM6 cells were set at different effector–target (E:T) ratios. The level of transduction in the CAR cells was set to 50% ± 10% for these experiments. After 24 h, the activation of Jurkat cells by CD69 and CD25 staining and secretion of IL-2 were analyzed. As shown in Figure 1A, no CD25 activation was observed in these conditions. Also, IL-2 levels were only very slightly increased in the 1:2 and 1:5 E:T ratios and below the reported limit of detection of the technique (7 pg/mL) (Figure 1B). On the other hand, CD69 showed a robust increase in its expression, which was specific to transduced cells (CAR) (Figure 1C,D). Therefore, a potency assay based on the quantification of T-cell activation by measuring CD69 expression was selected for further development.

### 3.2. Development of CD69-Based Potency as a Quality Control-Suitable Assay

CD69 expression level was determined in terms of MFI, in the CD3+ cell population, using the gating strategy shown in Appendix A. The specificity of the antibodies used (anti-CD69 and anti-CD3) was confirmed by using positive and negative cells, in the case of the anti-CD3 antibody, and blocking antibodies in the case of anti-CD69 (Appendix A).

To convert the assay to a quantitative assay, increase inter-assay precision and make the assay easily transferable to other equipment/laboratories, several conversions and normalizations were applied to the CD69 MFI value to obtain a final “CD69 activation index”. A summary of the conversions applied is shown in Appendix A. First, an internal calibrator was included in each assay. For this, a 4-point bead standard with a known number of fluorochrome molecules bound to the beads (MESF beads) was run in parallel to the samples using the same conditions, thereby creating a standard curve. Using the standard curve, “CD69 MFI” values were converted to “CD69 MESF” values (Appendix A). Second, a factor to normalize for variations in fluorochrome–antibody ratios for different antibody batches was introduced, converting “CD69 MESF” values to CD69-adjusted MESF (CD69 aMESF). Finally, as shown in Figure 1C,D, CD69 level of expression increased proportionally with the E:T ratio used, indicating good specificity and sensitivity of the assay. E:T ratio = 1:5 was selected for further development of the assay, since in this condition, a higher discrimination with basal cell activation was observed. Indeed, a basal level of activation of CAR-expressing Jurkat cells was also observed when compared to untransduced cells in the absence of target cells (E:T ratio = 1:0) (Figure 1D); therefore, a normalization of CD69 aMESF value with the CD69 aMESF observed in untransduced cells and with the basal condition (E:T ratio = 1:0) was applied, obtaining “CD69 activation index” as the final potency indicator. This normalization eliminates potential variability introduced by different basal cell activation levels in different experiments. “CD69 activation index” was calculated according to the following formula:$$\begin{array}{l} CD69\;activation\;index\\ =\frac{\left[CD69\;aMESF\;CAR\;E:T\;ratio=1:5\right]/\;\left[CD69\;aMESF\;UNT\;E:T\;ratio=1:5\right]}{\left[CD69\;aMESF\;CAR\;E:T\;ratio=1:0\right]\;/\;\left[CD69\;aMESF\;UNT\;E:T\;ratio=1:0\right]} \end{array}$$

Representative images of CD69 expression levels observed by flow cytometry using UNT and CAR-expressing Jurkat cells are shown in Appendix A.

### 3.3. CD69-Based Potency Assay Optimization

Once the endpoint of the assay was defined, the next step was to further optimize the assay conditions and test assay robustness. For this, we first analyzed how the percentage of cell transduction impacts the results obtained in the potency assay. We first transduced Jurkat cells with our CAR19-LVV at different multiplicity of infection (MOI) and analyzed the percentage of CAR-expressing cells 72 h later. As shown in Figure 2A,B, an increasing percentage of transduced cells was obtained when the MOI increased, as expected. A linear relationship was observed between 0 and 60% transduction. At higher MOIs (>60% CAR+ cells), a plateau was observed; due to the fact that in this condition, each cell is transduced with multiple viral particles, introducing higher variability in the results. Therefore, we aimed to define a transduction rate in the upper range of the linear curve (40–60%). We next analyzed how the transduction rate influenced the result of the potency test. As shown in Figure 2C, a linear relationship was observed between the percentage of transduced cells and the “CD69 Activation Index” obtained. This result indicated that adjustment of the percentage of transduced cells was necessary to obtain comparable results when measuring the potency of different LVV batches. To get a better understanding of how small variations in the percentage of transduced cells affect the measurement of potency, we conducted a second set of experiments, where CAR19-LVV-transduced cells at 60% transduction were mixed with untransduced cells to obtain a range of transduced cells from 60 to 40%, in 2.5% intervals. As shown in Figure 2D, a positive trend was observed again between the percentage of cell transduction and potency measurement; although within this range, significant differences were only observed between 40% and 50% transduction conditions. Based on these data, we defined a target transduction rate of 50% ± 7.5 for the potency assay.

Another critical parameter to optimize was the incubation time during E:T cell co-culture. To evaluate the optimal time point, different plates were set for each time point and processed for staining at the indicated times. CD69 is a rapid activation marker. Therefore, the selected time range was 6–30 h. As shown in Figure 2E, no statistically significant differences were observed in the CD69 activation index within this time range. These results positively reflected the robustness of the assay. A 24 h time point was selected for assay validation as this time point was better fitted in laboratory workflow.

### 3.4. Assay Validation

A scheme of the developed assay with the previously optimized conditions is shown in Figure 3A. This assay was validated for use as a potency assay for our CAR19-LVV batches, according to the principles described in the corresponding EMA guideline [31], and following the establishment of master and working cell banks for Jurkat cells, the evaluated main parameters included in assay validation were as follows: (i) precision, (ii) linearity and range, (iii) specificity and (iv) relative accuracy.

To evaluate the precision of the assay, intra-assay precision, inter-assay precision and intermediate precision were evaluated following recommendations reviewed in [32]. Inter-assay precision and intermediate precision were evaluated simultaneously. Three different samples were evaluated in triplicate, in four independent runs. Two of the runs were performed by “Operator 1” in “Instrument 1”, and the other two were performed by “Operator 2” in “Instrument 2”. Using this Efficient Design of Experiment (DOE) approach [33], all components of precision were evaluated in a total of 4 runs. The acceptance criterion for precision was set to ≤25% CV, as typically accepted for cell-based assays [34,35]. Results are presented in Table 1 and Table 2. As shown, for intra-assay precision, the mean %CV was calculated from a total of 12 samples/runs, with a value of 8.65%, well below the acceptance criterion. Similarly, inter-assay and intermediate precision were calculated for the three samples analyzed. %CV ranged from 15.96 to 20.51, again below the acceptance criterion.

The linearity and range of the assay were evaluated using APC MESF beads used as a standard. The range of the assay was assessed between the lowest and highest MESF values, where inter-assay precision and linearity acceptance criteria were met. The acceptance criterion for linearity was set at r^2^ ≥ 0.98. As shown in Table 3, linearity and range acceptance criteria were met for the whole range of the assay.

The specificity of the assay was evaluated using untransduced Jurkat cells, CARBCMA-LVV-transduced cells (as negative controls) and CAR19-LVV-transduced cells. CARBCMA-LVV contains a transgene encoding a CAR anti-BCMA protein. In this setting, it was used as a suitable negative control since NALM6 cells are negative for BCMA expression; therefore, no CD69 upregulation was expected. For this analysis, 4 CARBCMA-LVV batches, 26 CAR19-LVV batches and 26 untransduced Jurkat cells in independent experiments were used. As shown in Figure 3B, a statistical difference was observed when comparing CAR19-LVV Jurkat cells to untransduced cells and CABCMA-LVV, confirming that the assay is specific for CAR19-LVV samples. This result also shows that this assay is able to discriminate between Jurkat cells expressing a functional CAR19 molecule and untransduced cells (100% specificity).

Finally, the relative accuracy of the assay was also analyzed. As shown before, the level of CD69 expression directly correlated to the E:T ratio used in the co-culture (Figure 1D). A positive correlation was expected as a higher number of target cells in the co-culture (or a higher number of ligand molecules) resulted in a more robust T-cell activation. Therefore, relative accuracy was measured using five different E:T ratios ranging from 0 to 5. This result provided a significant correlation (Pearson’s test, *p* = 0.0047) with a linear equation [y = 216.1x + 1722]. Relative accuracy was calculated in terms of “%Recovery”, using the formula: “%Recovery = [(Observed value/expected value) x 100]”. For each point, the mean CD69 MFI of the three replicates was used as the observed value. The expected level of CD69 activation (MFI) at each E:T ratio used was calculated using its linear equation. Acceptance criterion for %Recovery for relative accuracy was set to 80 ≤ x ≤ 120. As shown in Table 4, %Recovery fell between 80% and 120% at all E:T ratios analyzed confirming the relative accuracy of the assay meets the typical standards for a quality control assay.

### 3.5. CD69-Based Potency Assay Adaptation to Other Antigen-Targeting CAR-LVV

Next, we wanted to test if this assay could be used as a platform assay, being easily adapted to other CAR-LVV targeting different antigens. Thus, we aimed at developing a potency test based on our already established CD69-activation antigen, for CARBCMA-LVV. The assay was performed in the exact same conditions as the potency test for CAR19-LVV, with the exception of the target cell line used, which was replaced by U266 cells. U266 is a cell line that was derived from an MM patient and expresses high levels of BCMA on its membrane. Similarly to the development of the CD69-based potency test for CAR19-LVV, we first conducted a series of experiments to confirm that the conditions were optimal for the BCMAh-LVV potency test. As shown in Appendix A, specific activation of Jurkat-CAR-expressing cells was observed, with an optimal ratio identified as E:T ratio = 1:5, as observed with CAR19-expressing cells. In terms of critical parameters’ testing, the optimal percentage of CAR-expressing cells was determined to be 52.5 ± 5% in this case (Appendix A). Having verified that the assay behaved similarly to the previously developed assay, we tested the potency of 12 CARBCMA-LVV batches produced, also including untransduced cells as negative controls (Figure 3C). Also in this case, we observe that the potency test is able to discriminate between Jurkat cells expressing a functional CARBCMA molecule and untransduced cells with 100% specificity. These results suggest that the assay can be easily adapted to other CAR-encoding LVVs designed against different antigens.

### 3.6. Correlation of CD69-Based Potency Assay and CAR-T Cell Cytotoxicity

T-cell activation is an early step in the CAR-T-cell-mediated cytotoxicity process. We next investigated the relationship between CD69 activation and CAR-T cell cytotoxicity. For that, we used primary T cells from B-ALL patients since Jurkat cells do not display full cytotoxic activity. Primary T cells were selected from apheresis products, activated, transduced with CAR19-LVV and then expanded for 7–9 days before conducting the assay. As shown previously (Figure 1D), CD69 is upregulated with an increasing number of target cells in the co-culture, allowing for more CAR molecules on the T-cell’s membrane to be bound to its ligand. However, to be able to assess a possible correlation between activation and cytotoxicity, the E:T ratio must be maintained constant within the conditions of the assay. Therefore, to obtain a wide range of activation/cytotoxicity values within the samples analyzed while having a constant E:T ratio, three different clones of NALM6-cells with increasing expression of CD19 and NALM6-CD19 KO cells were used as target cells [28]. CD19 expression levels in the NALM6 cell clones used are shown in Appendix A. Both T-cell activation, by CD69 expression, and T-cell cytotoxicity were evaluated simultaneously within a single assay as depicted in Figure 4A. Similar to the previous test, first, optimization studies were performed to select an optimal E:T ratio (Appendix A). In this case, E:T ratio = 1:2 was selected for the assay, since CD69 expression levels are higher in primary cells. Also, the E:T ratio = 1:2 was an optimal ratio to evaluate cytotoxicity simultaneously, according to our previous experience. Briefly, co-cultures of NALM6 cells and CAR-T cells were established in 96-well plates at E:T ratio = 1:2. Effector cell number was adjusted in terms of CAR+ cells instead of total T cells, thus eliminating a possible impact of different transduction efficiencies in the results of the assay. Wells corresponding to E:T ratio = 1:0 (effector cells only) and E:T ratio = 0:1 (target cells only) were also included to normalize each parameter, activation and cytotoxicity, respectively, with the corresponding basal level. Cells were cultured for 24 h and, after that, stained with CD3, CD69 and 7-AAD and analyzed using flow cytometry. In this case, “CD69 fold activation”, calculated according to the formula:$$\begin{array}{l} CD69\;fold\;activation\\ =\left[CD69\;aMESF\;CAR\;E:T\;ratio=1:2\right]/\left[CD69\;aMESF\;CAR\;E:T\;ratio=1:0\right] \end{array}$$
was used instead of “CD69 activation index”, since UT cells expanded in the same condition as CAR-T cells were not available for all donors. As shown in Figure 4B, both “CD69 fold activation” and “% Target cell killing” increased with increasing CD19 expression in the target cells, following an exponential-curve fit (r^2^ = 0.9872) (Figure 4C). These results demonstrate that the level of CD69 induction is a good indicator of CAR-T cell cytotoxicity.

### 3.7. CD69 Activation Assay in the Context of Autologous CAR-T Cells

Having established the “CD69 activation index” as a suitable potency test to be used as a routine QC test method for CAR19-LVV using the Jurkat cell line, and having also observed its positive correlation with cytotoxicity using primary T cells, we next wanted to explore if CD69 activation assay could be used as a quantitative surrogate potency test for autologous CAR-T cell products. To test this, 10 CAR-T cell products and 10 untransduced cells from different donors were subjected to the “CD69-fold activation” potency test, while evaluating cytotoxicity simultaneously (as a qualitative assay). As shown in Figure 4D,E, a clear separation in “CD69 fold activation” was observed between untransduced (UT) cells and CAR19-LVV-transduced cells (CAR-T), indicating that this assay is specific also in the context of autologous cell products. Moreover, cell cytotoxicity was evaluated within the same assay as an orthogonal qualitative test. As shown, no decrease in target viable cells (%Surviving target cells) was observed when co-culturing NALM6 with untransduced T cells (Mean ± SD = 108.9 ± 16). However, “%Surviving target cells” fell below 25% in all samples analyzed when co-culturing with CAR-T cells (Mean ± SD = 16.4 ± 7) (Figure 4D). Cell cytotoxicity assay data confirm the specificity of the “CD69 fold activation” assay.

To better understand how the “CD69 fold activation” assay may be influenced by different product attributes, the effect of product composition in terms of CD4/CD8 ratio and %CAR+ cells was investigated. Quality attributes of the products included in this study are presented in Appendix A. As shown in Appendix A, CD4 cells display higher levels of CD69 expression than CD8 cells, both in basal conditions and after co-culture with target cells. Since the “CD69 fold activation” represents CD69 activation over the product’s basal levels, no correlation is observed between CD4/CD8 ratio and “CD69 fold activation”, as expected (Appendix A). Therefore, CD4/CD8 product composition does not have an impact on the assay’s endpoint. A potential effect of the product’s %CAR+ cells on “CD69 fold activation” was also analyzed, even though the assay is adjusted by a number of CAR+ cells. As shown in Appendix A, no statistically significant correlation was reached between the two parameters. However, a positive trend towards higher CD69 fold activation obtained in products with a higher percentage of CAR+ cells is observed. This observation may be attributable to a higher number of CAR molecules/transduced cells in products with higher transduction efficiencies.

Finally, a comparison of CD69 expression levels to traditional IFNγ surrogate assay was also conducted by analyzing IFNγ levels in the supernatant of the co-cultures. As shown in Appendix A, induction of the respective markers was observed in both assays upon co-culture with target cells. However, we noted that absolute IFNγ levels (both in basal and after co-culture with NALM6 cells) showed a bigger dispersion than CD69 expression levels, as indicated by the CV value of the different sets of data (0.29 and 0.23 for CD69 vs. 0.99 and 0.81 for IFNγ). Results are shown in Appendix A and Appendix A. These data indicate that CD69 may be a more specific parameter than IFNγ to represent CAR-T cell functionality. All these data together reinforce the potential use of CD69-based assay as a surrogate potency test for autologous CAR-T cell products.

## 4. Discussion

LVV-based gene therapy products are widely used as a starting material in the manufacturing of ATMPs or as drug products. LVV-potency assessment remains one of the major challenges during product development and has been appointed critical for drug product approval by regulatory agencies. Assessment of vector potency integrates several aspects of its biology, including viral genome integrity, transduction efficiency, transgene expression and transgene function [22,36]. The last one necessarily encompasses the rest and, therefore, is usually favored and/or requested by regulatory agencies. Furthermore, the selected potency assay is required to be quantitative, fully validated and demonstrate an association with the drug’s MoA.

We have developed an in vitro potency assay to assess transgene function in a quantitative manner, using a flow-cytometry-based method. Assessment of the functionality of viral vectors and gene therapy products necessarily requires a cell-based assay. Cell-based assays are intrinsically more variable than molecular assays due to increased complexity and dynamism in the system. In order to minimize the variability of the assay, we decided to use a T-cell line as the transgene-expressing cells instead of primary T cells, where inevitably, changes in T-cell donors would have a great impact on the results of the assay. Among T-cell lines, Jurkat cells have been extensively used to study T-cell signaling and response after activation, also in the context of CAR-T cells [29,30]. Although Jurkat cells are not capable of a full cytotoxic response (release of IFNγ or cytolytic activity), we and others have shown that Jurkat cells are specifically activated and release IL-2 upon CAR binding to its antigen [29]. Therefore, Jurkat cells represent a good model to assess CAR-encoding LVV potency. Of note, the establishment of master and working cell banks for cell-based assays is critical to maintaining the assay’s performance and consistency across time.

The use of surrogate assays to assess CAR functionality is the preferred option for the vast majority of autologous CAR-T cell manufacturers. Among these assays, secretion of pro-inflammatory cytokines and T-cell activation, which are both strongly connected to CAR-mediated cytotoxicity [22,23,24,26], have been proposed by many, with cytokine secretion measurement being the most frequently implemented test even though no gold-standard assay has been defined. For CAR-encoding LVV-potency testing, we explored several options in the context of Jurkat cells being used as the effector cells, including two different activation markers (CD25 and CD69) and IL-2 secretion. Among the three, the CD69 increase in expression reunited the key characteristics for a suitable QC routine testing, including specificity, linearity and robustness, and we selected this assay for further validation. Furthermore, CD69 measured in terms of MFI, could be developed as a quantitative test.

Flow-cytometry-based methods are usually considered semi-quantitative and are not suitable options for potency testing. However, the use of MFI (instead of percentage of cells), the implementation of an internal standard curve constructed using MESF beads and the relative expression of MFI in terms of MESF values transform this assay into a relative quantitative assay that is suitable for potency testing [32,37,38,39]. Indeed, the implementation of these changes resulted in an assay that was successfully validated, including intermediate precision, with different operators and different instruments.

Correlation between T-cell activation (CD69 upregulation) and target cell death was observed in primary T cells. This analysis prompted us to evaluate the use of the CD69-based potency test for CAR-T cell products. Indeed, CD69 expression has been used as a diagnostic tool for T-cell function after HSCT [40] and has been proposed as a surrogate assay for CAR-T cells [26,41]. Our exploratory study showed that CD69 fold increase is a specific marker of CAR-mediated T-cell activation in an autologous CAR-T cell setting and could be implemented as a quantitative surrogate assay for CAR-T cell products. Of note, CAR-T cell cytotoxicity, as an in vitro test reflecting the drug’s MoA, can be simultaneously evaluated as a semi-quantitative assay with no additional effort. One major advantage of this assay over other more specialized assays is that no specific equipment is required. Also, compared to IFNγ release assay measurement, this assay shows less variability among autologous products, implying an improved specificity. High variability in IFNγ levels from batch to batch, in the context of autologous CAR-T cell products, has been one of the major criticisms of using IFNγ assay as a surrogate potency test [25]. Additionally, the proposed CD69-based assay is less laborious, less time-consuming and has a lower cost, which contributes to reducing the costs of these therapies, which is especially important in the context of academically developed products like ARI-0001.

In summary, we have shown that CD69-based potency assay is a suitable quantitative potency test that has been implemented for routine QC batch release of ARI-0001’s CAR19-LVV. The assay could easily be adapted to other CAR-encoding LVV or different vectors. Our data also indicate that the use of a CD69-based potency assay could be further explored as a surrogate-potency test in the context of autologous CAR-T cell products with the advantage of a reduced variability among autologous products and the capability of simultaneously evaluating CAR-T cell cytotoxicity.

## Figures and Tables

**Figure 1 pharmaceutics-17-00303-f001:**
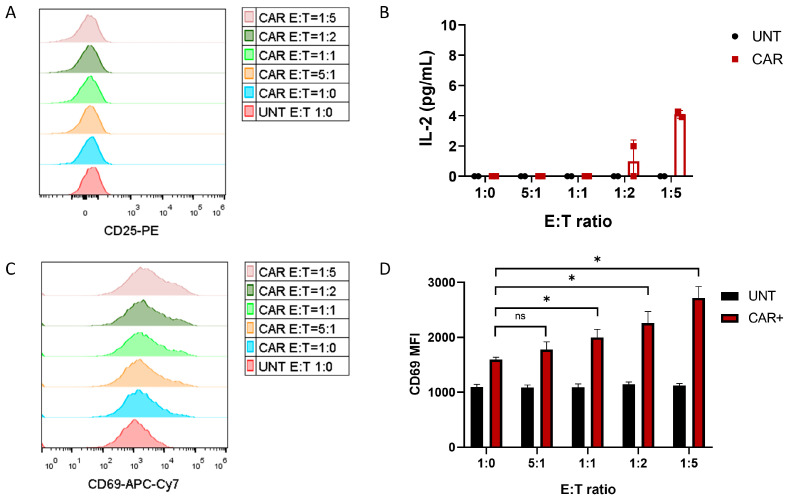
Readout selection for potency assay using CAR19-LVV-transduced Jurkat cells. (**A**) Jurkat cells expressing CAR19 co-cultured with or without NALM6 cells at the indicated E:T ratios. Histograms show MFI of CD25 staining in CD3+ cells at 24 h time point. (**B**) Presence of IL-2 in the supernatants of co-cultures used in (**A**). The mean of triplicates ±SD is shown. (**C**) Histogram plots of CD69 expression in CD3+ cells after 24 h of co-culture. (**D**) CD69 MFI quantification in co-cultures of untransduced and CAR19-expressing Jurkat cells and NALM6. The mean of triplicates ±SD is shown. “ns” indicates no statistical significance. “*” indicates *p* ≤ 0.05.

**Figure 2 pharmaceutics-17-00303-f002:**
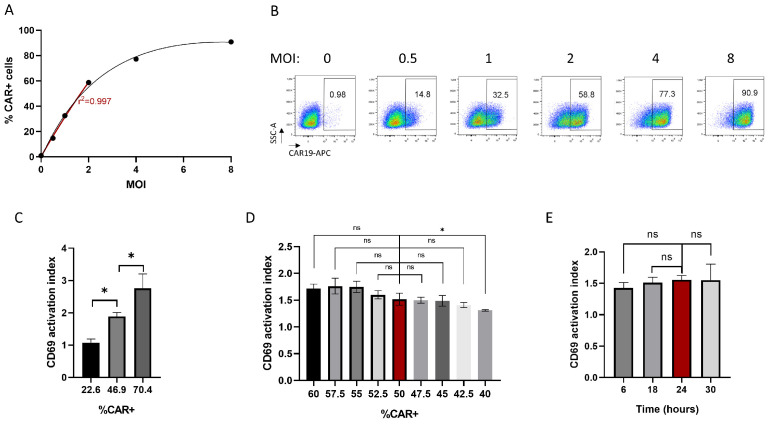
CD69-based potency assay optimization. (**A**) Correlation between the number of LVV particles used per cell (also known as multiplicity of infection (MOI)) and percentage of CAR-expressing cells at 72 h. The red line indicates a linear correlation. (**B**) Representative flow-cytometry images of CAR19-expressing Jurkat cells transduced at different MOIs. (**C**) CD69 activation index test performed using Jurkat cells displaying various percentages of CAR-expressing cells. Mean ± SD of triplicates is shown. (**D**) Same as (**C**) but using small intervals of CAR-expressing Jurkat cells. Mean ± SD is shown. (**E**) CD69 activation index test performed at the indicated time points after co-culture initiation. Mean ± SD is shown. “ns” indicates no statistical significance. “*” indicates *p* ≤ 0.05.

**Figure 3 pharmaceutics-17-00303-f003:**
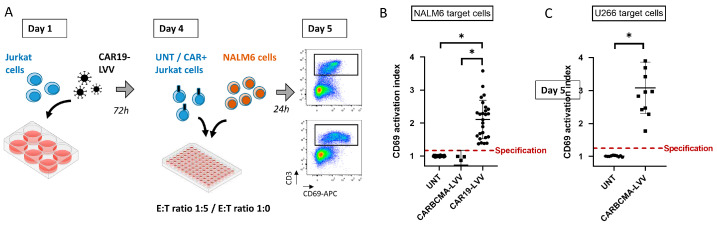
CD69-based potency assay test in routine LVV-batch analysis. (**A**) Diagram depicting the different steps of CD69-based potency test. (**B**) Results of potency test of 26 CAR19-LVV batches tested, 4 CARBCMA-LVV batches and 26 untransduced Jurkat cells. The dashed red line indicates the limit of the specification set for ARI-0001-LVV samples. Mean ± SD is shown. (**C**) Results of potency test of 12 CARBCMA-LVV batches and 10 untransduced Jurkat cells tested using U266 cells as target cells. The dashed red line indicates the limit of the specification set for CARBCMA-LVV samples. Mean ± SD is shown. “*” indicates *p* ≤ 0.05.

**Figure 4 pharmaceutics-17-00303-f004:**
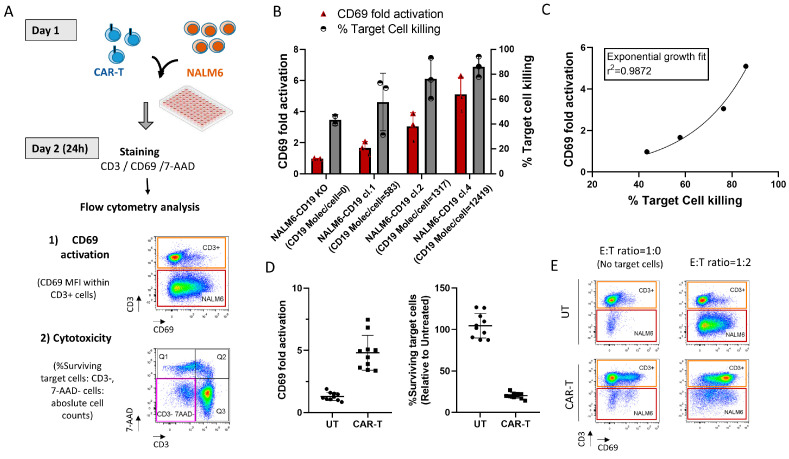
CD69 activation assay as a surrogate potency assay for autologous CAR-T cell products. (**A**) Diagram of CD69-based potency test applied to autologous CAR-T cell products. T-cell activation (CD69) and target cell killing or cytotoxicity are evaluated simultaneously. (**B**) Results of T-cell activation (CD69 fold activation) and target cell killing, using NALM6 clones with a variable number of CD19-molecules per cell as target cells. Results of triplicate experiments are shown. (**C**) Non-linear fit analysis between CD69 fold activation and %Target cell killing using data generated in (**B**). (**D**) Results of CD69-fold activation and cytotoxicity values obtained for 10 ARI-0001 batches analyzed. (**E**) Representative flow cytometry images of CD69 T-cell activation of patient batches analyzed in (**D**).

**Table 1 pharmaceutics-17-00303-t001:** Intra-assay precision evaluation.

	CD69 Activation Index											
	Sample 1				Sample 2				Sample 3			
Run	1	2	3	4	1	2	3	4	1	2	3	4
Replicate 1	2.48	2.10	2.06	1.66	2.51	2.26	3.04	1.93	2.40	2.50	2.94	2.00
Replicate 2	2.48	2.00	1.81	1.55	3.41	2.14	2.81	1.86	3.00	2.34	2.60	1.85
Replicate 3	2.23	2.04	1.62	1.62	2.86	2.05	2.21	1.75	2.23	2.21	2.59	1.67
Mean	2.40	2.05	1.83	1.61	2.93	2.15	2.69	1.85	2.54	2.35	2.71	1.84
SD	0.15	0.05	0.22	0.06	0.45	0.11	0.43	0.09	0.40	0.15	0.20	0.16
%CV	6.09%	2.50%	12.25%	3.67%	15.48%	4.94%	15.82%	4.88%	15.84%	6.22%	7.24%	8.87%
Mean %CV	8.65%											

**Table 2 pharmaceutics-17-00303-t002:** Inter-assay and intermediate precision.

	CD69 Activation Index				Mean	SD	%CV
Run	1	2	3	4			
Conditions	Op 1-Inst 1	Op 2-Inst 2	Op 1-Inst 1	Op 2-Inst 2			
Sample 1	2.40	2.05	1.83	1.61	1.97	0.34	17.08
Sample 2	2.93	2.15	2.69	1.85	2.41	0.49	20.51
Sample 3	2.54	2.35	2.71	1.84	2.36	0.38	15.96

**Table 3 pharmaceutics-17-00303-t003:** Linearity and range.

	MESF Value	MFI				Mean	SD	%CV
		Run 1	Run 2	Run 3	Run 4			
Bead 1	994	147	128	156	130	140.25	13.52	9.64%
Bead 2	4456	654	647	681	661	660.75	14.66	2.22%
Bead 3	24,312	3511	3640	3616	3777	3636	109.42	3.01%
Bead 4	73,490	11,090	11,161	11,528	11,826	11,401.25	342.09	3.00%
r^2^		0.9998	1	0.9997	0.9999			

**Table 4 pharmaceutics-17-00303-t004:** Relative accuracy.

E:T	Expected CD69 MFI	Observed CD69 MFI	% Recovery (80–120%)
0	1722	1595.67	92.66
0.2	1675.2	1777.67	106.12
1	1938.1	1998.33	103.11
2	2154.2	2262.33	105.02
5	2802.5	2718	96.98

## Data Availability

The authors confirm that the data supporting the findings of this study are available within the article and/or its Appendix A.

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
