# Peer review of "A Quantitative Approach to Potency Testing for Chimeric Antigen Receptor-Encoding Lentiviral Vectors and Autologous CAR-T Cell Products, Using Flow Cytometry"

_pharmaceutics, 2025, doi:10.3390/pharmaceutics17030303_

Round 1
Reviewer 1 Report
Comments and Suggestions for Authors
Overview
CAR-T-cell based therapeutics are paving their way towards clinical applications therefore it is of a great importance to develop functional assay for quantitative measurement of CAR-T potency.
Here authors developed a potency assay on the basis of flow cytometric CD69 expression measurement which reflect CAR-T cell activation upon binding to its target cells.
This test is useful for different CARs and autologous CAR-T products.
Comments
From the Introduction section (line 73 -74) it is not clear what are the other previously developed potency tests. Please specify and compare with the CD69 expression test in the discussion section.
Are there any other previous tests that were based on CD69 expression?
Acceptance criterion for precision was set to ≤ 25% CV. Please explain why this value was selected. Are there any other assays which relied on this value?
Other potency assay should be discussed in more details, obtained results should be compared to the published data.
Minor issues
Fig2- For clarity and consistency panels should be labeled as A, B, C in the first row and D, E in the second row.
Fig3B -Please label target cells as in Fig3C.
Fig4- How the number of CD19 molecules per cell was calculated?
Author Response
"Please see the attachment"

Reviewer 2 Report
Comments and Suggestions for Authors
This interesting paper by Mata et al. describes a relatively simple assay to assess the functionality of CAR lentiviral vectors using CD69 expression. This method may also be used to assess the potency of CAR-T products. The authors set up the method for CD19 lentiviral particles, but it can easily be adapted for other CAR products. While the method described is very relevant, there are several aspects of the manuscript that require clarification.
Specific comments:
1. As the method is designed to test a compound used in pharmaceutical manufacturing, guidelines available on analytical validation also apply on the method described. An example of such a guideline is the ICH Q2 guideline describing analytical validation. Reference should be made to this guideline and other relevant guidelines.
2. The authors explored CD25, IL2 and CD69 as potential candidates for the potency assay. It would be helpful to have some background on why these specific targets were selected. I reckon that plenty of other candidates are described in literature and have been considered. I would suggest to include some background on the selected candidates in the introduction.
3. The authors indicate that no gold standard currently exists to assess potency of lentiviral vector particles. They, however, indicate in the discussion that cytokines release is frequently used. From the discussion it is currently not clear, why the new method has advantage over this frequently used method. This aspect should be discussed.
4. The authors indicate that the IL-2 expression (Figure 1B) is near the limit of detection of the ELISA. It would be helpful to include the limit in the manuscript.
5. The authors indicate that the method is linear until a MOI of 2. For the production of CAR-T cells often higher MOIs are used. Could the authors commend on this potential mismatch? In addition, the authors indicate in the text that higher variability may be introduced at higher MOIs. In Figure 2A, however, the variability appears to missing.
6. The authors indicate that the acceptance criterion for precision was set at 25%. It is not clear from the manuscript what the basis of this value is. Please include a reference or rationale. Similarly, some other criteria are mentioned without reference. Please include these as well.
7. Autologous CAR-T products are a mixture of CD4, CD8 and some other T-cells. Did the authors investigate whether the composition of the CAR-T product has an effect on the expression of CD69?
8. The authors indicate that they normalized the number of T-cells in the assay based on CAR%. To interpret these results properly, it would be helpful for the readers to include the range of the CAR expression of the products in the manuscript. If differences between CAR expression is high, this would mean that the number of untransfected cells included in the assay is also highly variable. This could have an impact on the results of the potency assay.
9. The relationship between CD69 expression and target cell killing does not appear to be linear, but may be exponential (Figure 4C). Did the authors consider this option? In addition, what would the impact of an exponential increase be? Moreover, did the authors also find a relation between CD69 induction and target cell killing for the CAR-T products (Figure 4D)?
10. For the autologous CAR-T products, an E:T ratio of 1:2 was used. In the other experiments, an E:T ration 1:5 was used. Is this ratio of 1:2 correct?
11. Did the authors generate a master and working cell bank for the cells used in the assay. Would this be something to include or recommend?
12. Minor point: some sentences are rather long. The first paragraph of the introduction is one sentence of five lines. This is impressive! I would recommend to check the manuscript for too long sentences.
Author Response
"Please see the attachment"

Reviewer 3 Report
Comments and Suggestions for Authors
Dear authors,
Dear editors,
The article “A quantitative, flow-cytometry based method to assess functional potency of CAR-encoding lentiviral vectors and autologous CAR-T cell products” by Juan Jose Mata et al., aims to present a quantitative method for assessing CD69 expression for determining the CAR T potency. The authors present a validated in vitro cell based potency test, using a flow-cytometry method for their CAR T (CAR19-LVV). The test measured CD69 potency allowing also the evaluation of cytotoxicity.
The article looks great, and the authors managed to present in detail all the workflow involving the flow cytometry method for assessing the CAR T potency.
Please find my point-by-point observations:
The introduction section contains all the background information needed to understand the aims and scope of this study, the authors described the LVV applications and presented the CAR T cells use in clinical practice. Also, several clinical and preclinical studies are presented and finally the authors described the ARI-001, and the CD69 potency test.
For the introduction section I have no further observations.
Results section:
2.1. Selection of a potency assay for ARI-001 LV vector
- The results are clearly described and figure 1 has all the information needed.
- Please add in the figure legend how many replicates were used.
2.2. Development of CD69-based potency as a quality control-suitable assay
- no comments for this chapter
2.3. CD69-based potency assay optimization
- please mark in the text or in figure legend for 2B – what does 1 MOI represents, to make the MOI scaling easier to understand by readers.
- no other remarks, the results look great.
2.4. Assay validation
- The results present a clear validation of the method. No further remarks.
2.5. CD69-based potency assay adaptation to other antigen-targeting CAR-LVV and 2.6. Correlation of CD69-based potency assay and CAR-T cell cytotoxicity
- The results are well presented and the CD69 activation is related to the killing potency of the CAR Ts.
- figure 4 is complete, however, the gating text in yellow is barely visible, I recommend changing this to make it visible.
2.7 CD69 activation assay in the context of autologous CAR-T cells
- no further comments.
Discussion section:
- The authors have demonstrated that the CD69 based potency assay can be used in routine , the method being easily adaptable for CARs with LVV or even different vectors. Also the CD69 based potency assay can be explored for a surrogate potency assay in autologous CAR T products.
Materials and methods
4.3. Jurkat cell transduction
- There is mentioned the MOI ranging from 0.25 to 4, however in figure 2 you have a range up to 8. Please check this data, there might be a typo.
Reference list
- No issues detected
All being said, the paper needs minor revisions, and I would like to congratulate the authors on their work. The manuscript could be one of interest for readers and the presented method can be used by other research groups to increase the quality of their research.
Author Response
"Please see the attachment"

Reviewer 4 Report
Comments and Suggestions for Authors
The manuscript titled "A Quantitative, Flow-Cytometry-Based Method to Assess Functional Potency of CAR-Encoding Lentiviral Vectors and Autologous CAR-T Cell Products" represents a noteworthy contribution to the field of immunotherapy, specifically in the evaluation of CAR-T cell products. The authors aim to establish a quantitative assay for measuring the functional potency of CAR19-LVV, with potential applications for other CAR-encoding vectors. While this innovative approach offers valuable insights into CAR-T cell potency, addressing the following considerations could significantly strengthen the manuscript's credibility and applicability:
1- The abstract is quite dense and should more concise. Key findings should be highlighted more clearly.
2: While the manuscript claims that CD69 expression is a specific indicator of CAR functionality, it does not fully address potential cross-reactivity or alternative pathways that may influence CD69 expression.
3: The manuscript discusses optimizing transduction rates but does not provide comprehensive data on how variations in transduction efficiency affect assay outcomes across different experiments.
4: The manuscript mentions statistical significance (e.g., p-values) but does not provide detailed information on the statistical tests used or their appropriateness for the data type.
5: It is important to evaluate conclusions made from CD69 expression levels in light of other established markers of T-cell activation and cytotoxicity.
6: While the authors suggest that their assay can be adapted for other CAR constructs, they do not provide sufficient evidence or examples to support this claim.
7: Authors should compare their assay with existing potency assays to provide insights into its relative advantages or limitations.
8: The title may be overly complex due to jargon and abbreviations. A more straightforward title, such as "Optimizing CAR-T Cell Efficacy: A Quantitative Approach to Potency Testing Using Flow Cytometry," could improve accessibility
9: While authors noted that transduction rates significantly affect the potency assay results, the exact mechanisms behind this relationship remain unclear.
10- The assay's dynamic range should be defined, particularly at low E:T ratios where sensitivity may be compromised.
11: The manuscript primarily focuses on static measurements of potency. Longitudinal studies assessing how CAR-T cell functionality changes over time post-infusion would provide insights into the durability of the therapeutic effect.
Author Response
"Please see the attachment"

Round 2
Reviewer 2 Report
Comments and Suggestions for Authors
The authors have appropriatly addressed the issues raised. Well done!
Reviewer 4 Report
Comments and Suggestions for Authors
The authors have responded to all the reviewer comments and the manuscript has been significantly improved and now may be publishable
Comments on the Quality of English Language
The English could be improved to more clearly express the research